# Wireless Strain Gauge for Monitoring Bituminous Pavements

Camille Gillot [1,†], Benoit Picoux [1,*,†], Philippe Reynaud [1,†], Debora Cardoso da Silva [1,†], Ndrianary Rakotovao-Ravahatra [1,†], Noël Feix [2,†] and Christophe Petit [1,†]

1 GC2D Laboratory, Limoges University, 5, Rue du 9 Juin 1944, 19000 Tulle, France; camille.gillot@insa-lyon.fr (C.G.); debora.cardoso_da_silva@unilim.fr (D.C.d.S.); ndrianary.rakotovao-ravahatra@unilim.fr (N.R.-R.); christophe.petit@unilim.fr (C.P.)
2 XLIM Laboratory, Limoges University, 7 Rue Jules Vallés, 19100 Brive la Gaillarde, France; noel.feix@unilim.fr
* Correspondence: benoit.picoux@unilim.fr
† These authors contributed equally to this work.

**Abstract:** This paper introduces the implementation of a new device for measuring deformations at the surface layers of bituminous pavement. Using wireless technology, rechargeable remotely, low cost, and easily positioned in a layer by coring after pavement construction, this sensor makes it possible to obtain measurements of the deformation when a vehicle passes by. The development of the wireless sensor is presented as well as its advantages and limitations. It was then tested in the laboratory under a hydraulic press and in situ using a full-scale test of the mobile load simulator (MLS10 type). This system allows simple measurement, gives reliable results, and could be a useful device for the structural monitoring of pavement structures.

**Keywords:** strain gauge; wireless sensor; monitoring; instrumented pavement

## 1. Introduction

Whether on in situ road sites for the monitoring of works, behavior, or fatigue tests of pavement in the laboratory, or to validate the calculation of the methods of pavement design, the instrumentation of the layers of pavement is increasingly used. Among the most widely used sensors, strain gauges make it possible to measure longitudinal or transverse deformations in the first layers of pavement. They are often placed in the treated layers (bituminous or treated with hydraulic binders). Many works concern this type of instrumentation. For example, in the USA, Swett et al. [1] installed 12 strain gauges in the asphalt layer and 4 strain gauges in the ground in a section of roadway in the state of Maine. This study proved that the installation of strain gauges in asphalt concrete is very delicate. Indeed, among the 12 sensors, 3 of them did not survive the road construction phase. Other projects have used strain gauges like Al-Qadi et al. [2]. In this project, H-shaped strain gauges and vibrating wire strain gauges were installed on a section named Virginia Smart Road. Xue et al. [3] worked with 10 longitudinal and 4 transverse strain gauges on an experimental section of the Virginia State Road. In Canada, Maadani et al. [4] developed an experimental section in Ottawa. The asphalt strains were measured using Dynatest (Gainesville, FL, USA) strain gauges. These gauges were positioned within the asphalt concrete (AC) layer to measure the strain in the transverse and longitudinal directions with respect to the traffic flow. The gauges were installed near the bottom of the lower AC layer. However, to ensure that the measurements obtained are correct, the authors specified certain conditions of implementation. In a section in Louisiana, Elseifi et al. [5] showed that the survivability of the gauges was acceptable because none of the gauges failed during the installation. After construction, the longitudinal strain measurements were constant up to approximately 350,000 passes. The signals then seemed noisier but the maximum response could still be easily extracted. In China, Ai et al. [6] have set up instrumented experimental sections. Each section was instrumented, with four strain gauges having two

different orientations concerning the direction of traffic. In France, Gabori et al. [7] have set up groups of three strain gauges (vertical, longitudinal, and transverse gauges) at different levels in the bituminous layers. Duong et al. [8] used seven strain gauges. The longitudinal gauges at the base of the high-modulus asphalt concrete were damaged during construction (probably by the effect of the intense compaction of this material), and their measurements could not be used. Pouteau et al. [9] set up, on an instrumented site in the Yvelines (France), 24 deformation sensors positioned at the base of the layer of bitumen. These deformation sensors are connected to a new generation of self-powered acquisition boxes developed especially for this application. Data analysis is performed on a remote data server, using a specially developed software suite. Also in France, Oubadou et al. [10] implanted different types of deformation measurements in an experimental pavement pit to study the behavior of the structure under traffic simulator MLS10 tests (fiber optic network, longitudinal, and transverse deformation gauges under the pavement surface). Grellet et al. [11] developed an instrumented pavement core also equipped with optical fiber strain gauges. This was the first technique with set-up after the construction of the asphalt concrete layer.

All the previously cited studies highlight that to ensure sufficient anchoring in the layers of bituminous materials, the gauges are most often installed during construction. These must be robust enough to withstand the stresses of compaction and the high temperatures of the installation process. To do this, these are installed on resin or aluminum supports or glued on a thin strip protected by a metal tube and layers of insulating material. To avoid damages during installation or to install sensors after construction, the gauges are sometimes carried over the cores of material extracted after construction and put back into place by gluing. Whether placed before or after construction, these gauges require wiring that is often difficult to make and maintain over time. Mechanical and/or thermal breaking may occur during the first installation of the sensor, as this requires careful cable insulation. A small transverse trench must be made and then covered with asphalt. This manipulation undeniably affects the uniformity of the surface. In both cases, a connection to a frame grabber is needed.

With the progress made in recent years on wave transmission technologies, wireless sensors are increasingly used in the monitoring of engineering structures, such as bridges [12], buildings, railways [13,14], or road structures [3,15]. Many articles deal with their use in the dynamic and vibration fields [16–18] as well as in fatigue or damage problems [19,20], detection of the presence of humidity [21], or even vehicle detection and counting [22,23]. Di Graziano et al. [24] conducted a state-of-the-art survey on the structural condition monitoring of bituminous pavements using smart sensor networks. The authors collected the relevant literature on wireless sensor networks and pavement monitoring. This study delved into the essential aspects of networks designed for pavement monitoring with a focus on damage detection. These fundamental characteristics include energy supply (battery power, energy harvesting techniques, or energy efficient designs to ensure operational longevity), detection methods (accelerometers, strain gauges, or acoustic sensors), network architecture (sensors, communication protocols, data aggregation, etc.), but also performance validation procedures (laboratory tests, field experiments, simulation studies, comparison with existing systems). Pouteau et al. [9] and Wang et al. [25] have set up measurement or detection systems whose transmission is wireless. The deformation and temperature sensors are connected to self-powered acquisition units. Other researchers, like Alavi et al. [26] or Xiao et al. [27], proposed smart sensing technology based on the use of self-powered piezoelectric sensors that could be distributed in the pavement for continuous monitoring. The approach proposed by Alavi et al. [28] for the health monitoring of pavement systems involves a self-powered sensing system for the evaluation of damage detection performance. Numerical simulations are used to model the behavior of the pavement system under different conditions and to assess the effectiveness of the approach in detecting damage. Experimental studies were performed by implementing the wireless sensor pavement in pavement cores and testing its performance under various scenarios, as well, the damage localization and quantification are investigated and their methodology is based on relative damage. It is therefore not necessary to directly measure

the absolute deformation of the pavement. Lajnef et al. [29] highlighted the importance of continuous, long-term monitoring for pavement structures and presented a novel approach involving self-powered wireless sensors and advanced data analysis techniques. The authors present the feasibility and effectiveness of the proposed monitoring system; however, they also discuss the limitations and challenges encountered during the development and validation process, which may include issues related to sensor accuracy, data transmission, or computational complexities in strain distribution estimation. Rhimi et al. [30] present the development of a detection system capable of being applied to the long-term monitoring of roadway structures and determining damage and their lifespan. The module consists of a miniaturized, battery-free system with a wireless piezoelectric sensor capable of detecting the long-term deformation history of the pavement structure. The authors also studied the feasibility of integrating the sensor into the roadway, the survival of the sensor under construction conditions, and its longevity.

Aware of all these difficulties, in this paper we propose to produce and test a novel miniaturized wireless strain sensor that we instrumented on a pavement core. This pavement core is then re-established in the structure by gluing. The main advantages of this device are that it is post-instrumented (without fear of destroying the sensor during pavement construction), wireless, with relatively reduced size, low cost, and low energy. This could be a useful device for monitoring pavement evolution. This paper is organized as follows: Section 2 presents the electronic design of the sensor, Section 3 provides some details on sensor installation in the pavement surface layer, Section 4 shows the validation of the measurements process in laboratory and the in situ sensor test with mobile load simulator, Section 5 presents some in situ results and discusses results and improvements.

## 2. Electronic Design of the Sensor

Figures 1 and 2 show the synoptic and the gauge wiring diagram. For an application in bituminous materials, we choose a large gauge (type C2A-06-20CLW-120 from Micro-Measurements (Wendell, NC, USA) [31]). The strain acts on the gauges placed in the test body. The strain gauge conditioning circuit shapes the signal for analog-to-digital conversion. A 10-bit analog-to-digital converter (ADC) is embedded in the microcontroller unit (MCU). It can convert two channels with a 1 kHz frequency sampling. The MCU also contains a thermally compensated voltage reference in order to calibrate the ADC. The measured data are translated into UART (an asynchronous serial communication protocol) and sent to a data grabber via a Bluetooth transceiver. In our case, data grabbers can be popular devices, like a laptop or smartphone, which is very convenient for in situ testing.

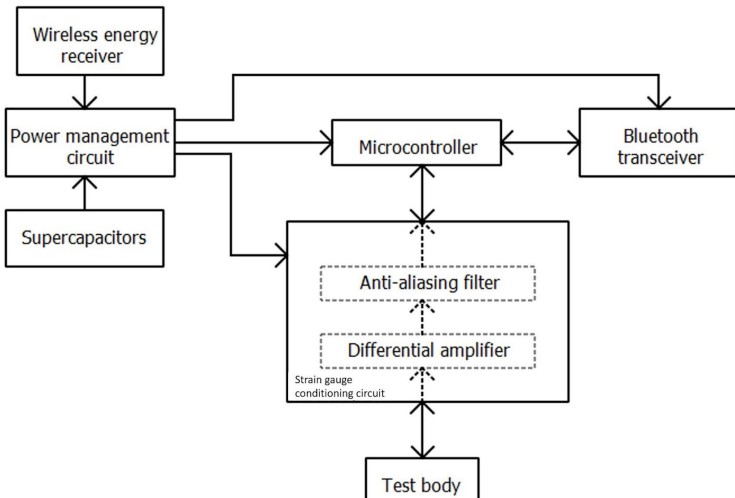

**Figure 1.** Synoptic.

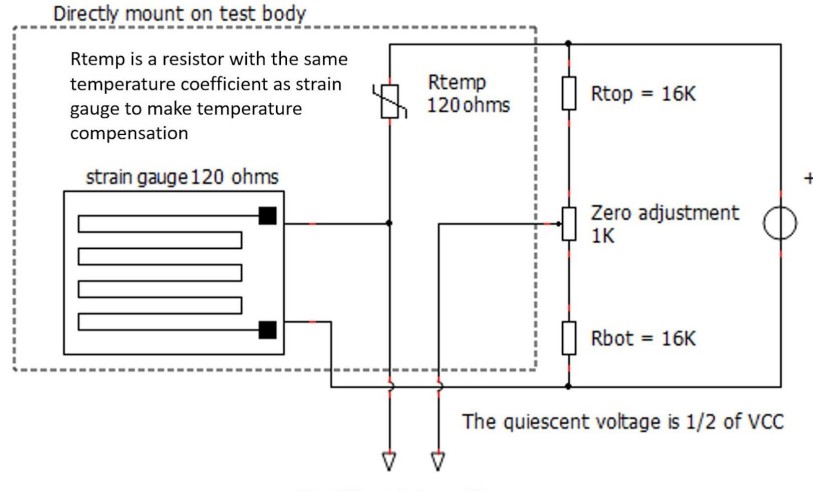

**Figure 2.** Gauge wiring diagram.

The circuit is equipped with a supercapacitor, and supercapacitors have several advantages over traditional batteries. At −40 °C, capacitance decreases around 10% for supercapacitors without any increase in leakage current. For lithium-ion batteries, it is about 20% at −20 °C. They are also able to withstand a greater amount of charge and discharge cycles than batteries, which can lead to a higher reliability for the device. Thus, power management circuit and supercapacitors provide energy for a duration of 5 min of measurement. The wireless energy receiver uses the charge of the supercapacitors when the measurement circuit is turned off. This also uses the same technology as a phone wireless charger. Supercapacitors are charged at constant current (approximately 500 mA) to avoid an inrush current and improve service life. It is considered that the discharge is also performed at a constant current (approximately 120 mA). Supercapacitors are oversized by a factor of 1.5 to consider aging, self-discharge, and losses. To operate correctly, the circuit must be powered by a voltage between 3.3 V and 5 V. Supercapacitors are also able to handle higher current loads than batteries. This is important given that for a short measurement period, the sizing characteristic for batteries is the current capability rather than the energy capacity. This is conducted to the oversized energy storage. With supercapacitors, the volume required for energy storage is reduced if charging is possible between two measurements.

The power management circuit has the following three functions: wake up the sensor, charge the supercapacitors, and supply the Bluetooth transceiver. To wake up the sensor, we use a reed switch, which is activated wirelessly with a magnet placed on the pavement when waking up the sensor is needed. The main advantage of this technique is zero energy consumption when the device is off. The charge is stopped using overvoltage detection. The MCU is used to drive the measurement channels and convert analog-to-digital and UART translation. The data logger (smartphone or laptop) receives raw data. The objective of such a setup is to reduce the MCU's tasks to improve the sampling frequency (event-based programming with hardware interrupts). At the end of measurement, the circuit is turned off by a software instruction that interacts with the power management circuit to avoid leakage or quiescent current when the device is off.

The choice of an asymmetrical Wheatstone bridge (see Figure 2) with four unequal impedances is made to save energy. The Rtemp and the gauge form a divider bridge that delivers a third of the supply voltage (here VCC = 2.5 V). A third of the voltage is chosen to ensure correct biasing even for the negative part of the expected signal. Rtop, Rbot, and the adjustment potentiometer make it possible to adjust the voltage difference in the two dividing bridges so that the output of a differential amplifier corresponds to one-third of the full-scale voltage (FSV) of the ADC. The zero adjustment potentiometer is electronically tuned by the microcontroller at each start-up. To improve the precision of

the zero adjustment, the program includes a zero-strain voltage measure to set precisely at zero the measured strain. For gauge conditioning, we use a thermistor mounted on the test body to compensate for the slope change due to the temperature variation. The divider bridge formed by the gauge and the thermistor delivers half of VCC at zero strains. The signal is digitized by the embedded ADC of the MCU. Also, the MCU converts data from the ADC to the UART protocol to interface with the Bluetooth transceiver. We use a fast Fourier transform (FFT) of the expected signal to estimate the needed sampling frequency. As we can see in Figure 3, the frequencies contained in the signal do not exceed 10 Hz. To keep a high signal–noise ratio (SNR) and a nice curve shape, we choose a minimum sampling frequency of 200 Hz. To improve the quality of the analog-to-digital conversion, we place an anti-aliasing filter. The cut-off frequency is set by the maximum sampling frequency divided by two to avoid spectrum aliasing. In our case, if the sampling frequency is 1000 Hz, the cut-off frequency of the anti-aliasing filter should be set at 500 Hz. The use of an anti-aliasing filter prior to analog-to-digital conversion improves the digital signal quality by reducing quantization noise and preventing spectrum aliasing. As a result of this design, the sensor resolution is 1 µm/m, its span is −250 to 750 µm/m with an accuracy of 5% and a sampling frequency up to 1 kHz.

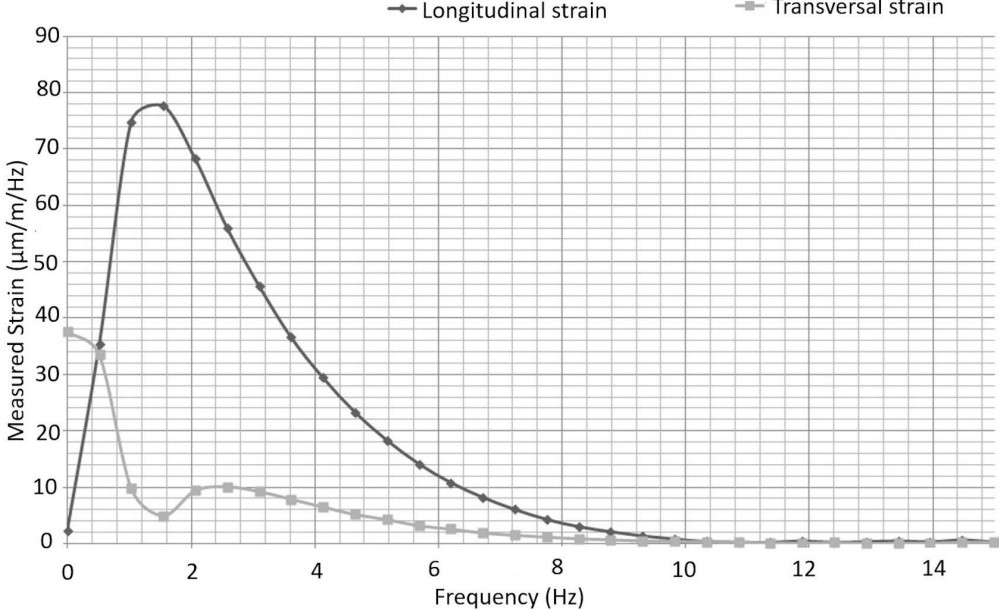

**Figure 3.** FFT expected signals.

## 3. Installation of the Sensor in the Pavement Surface Layer

Two core samples were taken at a distance of 2.5 m on the pavement. The first has a diameter of 108 mm and provides us with a core with a diameter of 96 mm. The second has a diameter of 97 mm and provides us with a core with a diameter of 90 mm. The latter is kept for laboratory measurements. This method is essential because it guarantees that the instrumented core has the same module as the rest of the pavement and, therefore, good continuity of the deformation field. The first is instrumented and then glued in the 97 mm diameter hole using epoxy glue. We chose epoxy glue, which has a tensile and flexural modulus near the grave bitumen layer one. Other technical constraints, like glass transition temperature, polymerization time, and positioning time, are considered. For the first laboratory test under hydraulic press, the test body is equipped with the following two gauges: one for longitudinal strain and another for transversal strain (Figure 4c). We chose a long strain gauge to measure the average strain of the test body. Their instrumentation involves several steps:

- The surface of the test body where the gauges will be glued is grounded and a groove is made on the edge of the cylinder for the passage of the cables (Figure 4a);

- A layer of hot bitumen is applied to fill the gaps. In order to perfect the surface condition, an epoxy resin is applied and then sanded (Figure 4b);
- The gauges are glued using a glue intended for this purpose (Figure 4c). At this point, the thermal compensation resistors and the Wheatstone bridge are wired;
- A PVC (polyvinyl chloride) strapping and a protective paste against humidity is affixed (Figure 4d).

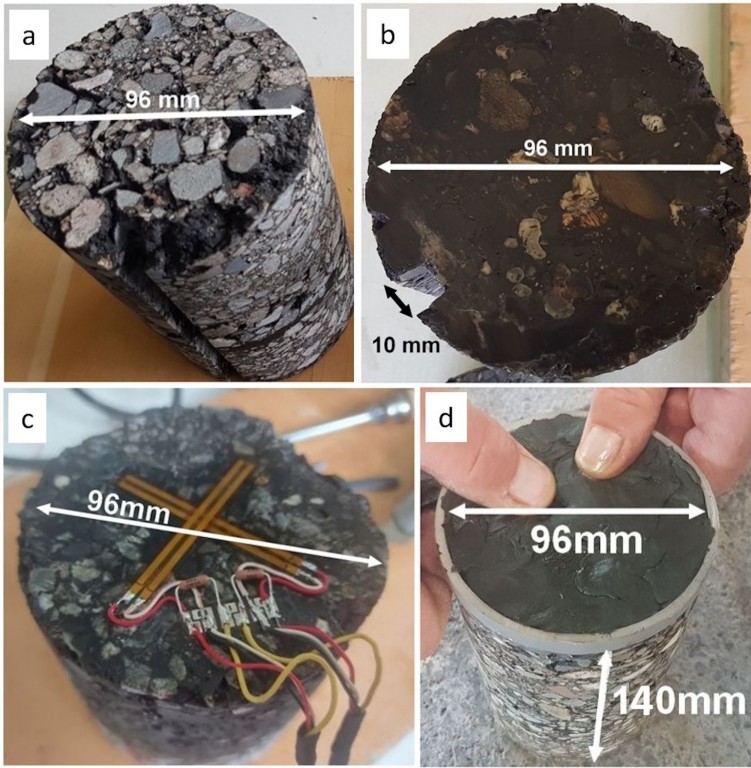

**Figure 4.** Preparation of the test core. (**a**): Machined test core; (**b**): test core before gauges gluing; (**c**): two perpendicular gauges; (**d**): finished test core.

With machining, the bottom of the drilling core is removed and a void appears between the bottom of the hole and the instrumented core. This is simply filled with sand. To ensure the correct alignment, a line parallel to the longitudinal gauge axis is drawn on the top of the test body. During gluing, the core is placed so that the top line is aligned with the pavement direction. After the installation of the core in the pavement layer, a groove is made between the two core holes. This will receive the cables of the classic gauge system (Wheatstone bridge) used for comparison with our wireless sensor. The remaining hole receives a PVC box containing the sockets for connecting the instrumented core to the electronics. This box is waterproof and is flush with the surface of the road. Figure 5 shows a schematic of the test setup. During measurements, the center axis of the MLS10 twinning is positioned at 550 mm from the measuring instrument using a system of marks to ensure the alignment of the longitudinal axis of the gauge with that of the wheels.

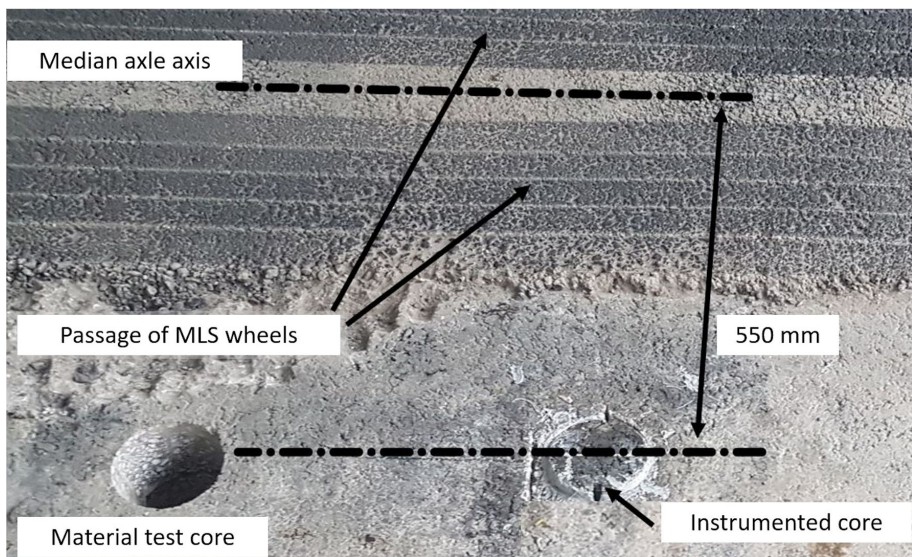

**Figure 5.** Sensor installation in the pavement.

## 4. Validation of Measurement Process

To check each part of the sensor separately, we proceed as follows:

### 4.1. Diametral Compression of a PVC Test Body Instrumented with One Strain Gauge

We made a test body out of PVC (Figure 6, right) and compressed it (Figure 6, left) to check the linearity of the measurement and validate the electronic section of the measurement channel. A Zwick press with diametrical shims is used to load the test body and obtain a homogeneous strain; also, the diametrical compression is used to calibrate the measurement channel. This method ensures that the measurement channel is providing accurate and consistent readings. Indeed, this procedure is necessary because the uncertainties about the component values and analog offset compensation are too important and the channel measurement can be saturated without mechanical load on the pavement. As can be seen in Figure 7, the response of the gauge is perfectly linear. An offset of approximately 500 N appears between the loading and unloading curves. This shift is due to the stress relaxation and creep deformation of the PVC core.

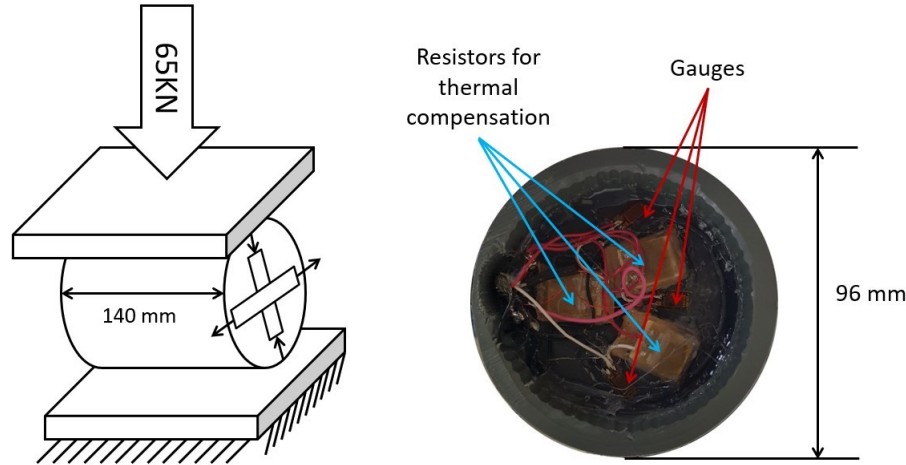

**Figure 6.** (**Left**): Diametric compression schematic. (**Right**): View of the PVC core.

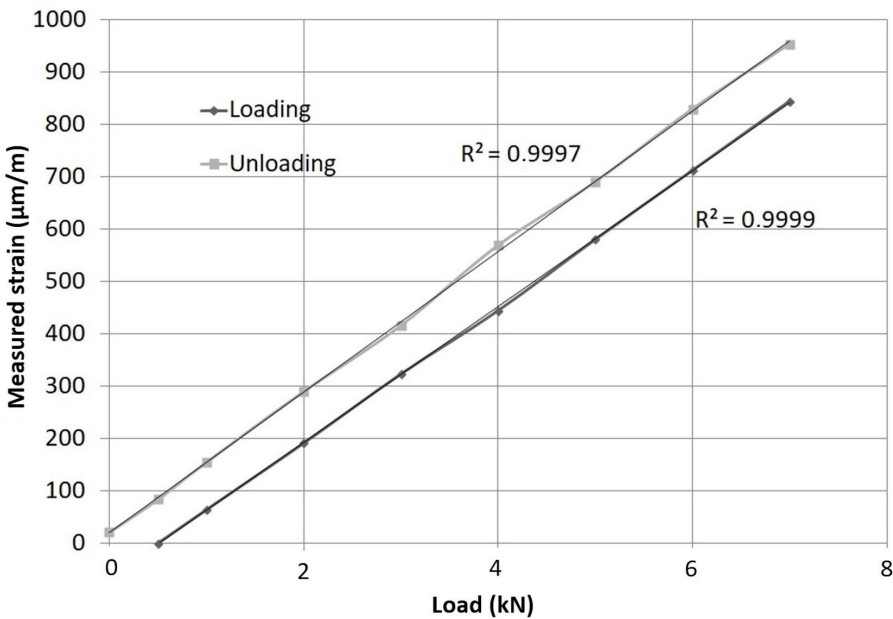

**Figure 7.** Sensor response with a PVC test body under a loading/unloading cycle.

To evaluate the nonlinearity error due to the conditioning circuit based on the asymmetrical Wheatstone bridge, we chose to calculate the worst-case operating condition of the measurement chain. Table 1 resumes the relative error between real and measured strain. The real strain is an imposed displacement range. From an imposed displacement, the corresponding resistance of the gauge is determined thanks to the formulas from Freire [32]. The measured strain is computed from Rg, and takes into account the nonlinear behavior of the conditioning circuit and ADC. The relative deviation is the absolute ratio between the real and measured strain.

**Table 1.** Relative deviation between real and measured strain.

| Real Strain (µm/m) | Rg (Ohm) | Measured Strain (µm/m) | Rel. Deviation (%) |
| --- | --- | --- | --- |
| −200 | 119.95 | −200.03 | 0.02 |
| −100 | 119.97 | −98.81 | 1.19 |
| −50 | 119.99 | −50.61 | 1.22 |
| 0 | 120.00 | Indefinite | ∞ |
| 50 | 120.01 | 50.61 | 1.22 |
| 100 | 120.03 | 101.22 | 1.22 |
| 200 | 120.05 | 200.03 | 0.02 |
| 300 | 120.08 | 301.25 | 0.42 |
| 400 | 120.10 | 400.06 | 0.02 |
| 500 | 120.13 | 501.28 | 1.28 |
| 600 | 120.15 | 600.09 | 0.02 |
| 700 | 120.18 | 701.31 | 0.19 |

The relative error is infinite at 0 µm/m, and it is mainly due to the conditioning. In a perfectly balanced Wheatstone bridge, the voltage difference between the two outputs would be zero at zero strains. However, in practice, due to imperfections in the bridge and thermal noise, a small voltage difference will always be present, even at zero strains. As a result, the SNR is 0 (0 in linear and −∞ in dB) when the real strain is 0. This error is typically negligible for large strains but may become more significant at lower strains.

### 4.2. Diametral Compression of a PVC Test Body Instrumented with Two Perpendicular Gauges

Figure 8 shows that the two gauges (longitudinal and transverse) have a linear behavior proportional to the force applied to the surface of the core; however, the absolute slope coefficient values are slightly different. This difference is due to the gauge factor and manufacturing dispersion of conditioning electronic circuits. Moreover, the symmetric axle between the two gauges is probably imperfectly aligned with the press axle. As a result, the strength is not perfectly distributed. This offset will be removed during data processing.

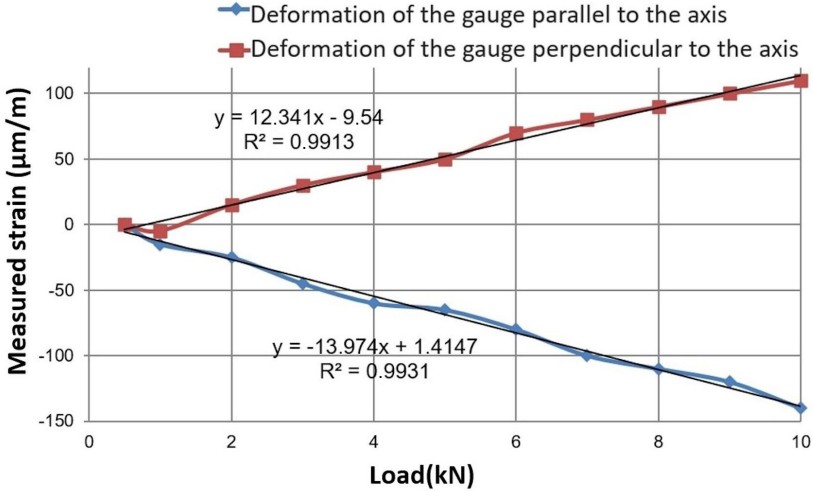

**Figure 8.** Response of the gauge against parallel and perpendicular axle (see Figure 4c).

### 4.3. Indoor Experiment on Instrumented Pavement Core with Two Parallel Gauges

To validate the gluing technique and the test body, we made a first test for our indoor experimental pavement [10]. The test body is instrumented with the protocol already presented and glued in the pavement, but the gauges are now placed parallel to each other (see Figure 9, left). One gauge is wired to the wireless measurement channel and the other is wired to a traditional measurement channel (this traditional measurement used for comparison between gauges requires the presence of a small trench up to the acquisition system, Figure 9, right). Figure 10 compares the measurements between innovative and reference methods. The small gap between both curves is probably due to the use of two gauges. Indeed, one gauge is slightly further from the load than the other. As a result, the expected deflection is slightly different. After milling the surface course, we found that the glue joint is thin as expected. The glue joint is regular except close to the cable passage (Figure 9, right).

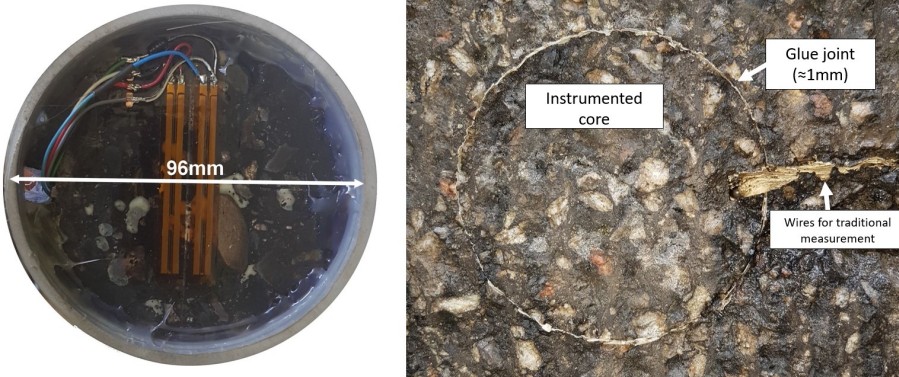

**Figure 9.** (**Left**): Test core with two parallel gauges (wireless and Wheatstone bridge), (**Right**): instrumented core placed in the pavement.

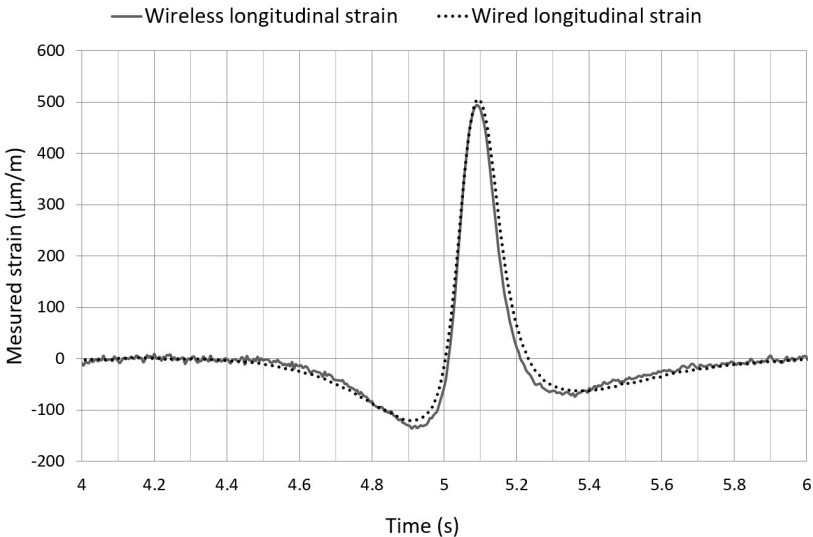

**Figure 10.** Comparison between innovative wireless method and wired reference method.

### 4.4. Outdoor Test with Real Load Condition and Comparison with Numerical Result

We used a mobile load simulator (MLS10) to test our system with a real moving load. This device involves simulating real-world traffic conditions on a full-scale section of pavement. This type of testing allows the evaluation of the performance of new pavement materials and technologies under realistic conditions, and can provide valuable information about the durability, longevity, and sustainability of the pavements. The MLS applies 65 kN loads on tandem axles (32.5 kN per axle) to the pavement surface at an 8 km per hour speed [11]. Figure 11 introduces the pavement structure and the MLS measurements. The studied pavement is a classic structure of the roads of the French network. The material of the wearing course is composed of 60 mm of asphalt concrete. The second layer is composed of a grave bitumen (GB3) 80 mm thick, then 3 layers of unbound granular material (UGM) and granitic sand. The mechanical parameters of the structure are those given by the pavement designer and have been verified by numerous FWD tests beforehand.

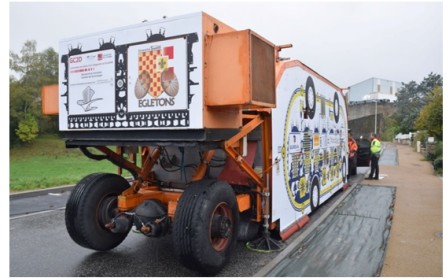

| Material | Young's Modulus (MPa) | Thickness (mm) |
|---|---|---|
| Asphalt Concrete (AC) | 7000 | 60 |
| Gravel stabilised with Bitumen (GB) | 11500 | 80 |
| Unbounded Granular Material (UGM) | 80 | 850 |
| Granitic Sand (GS) | 20 | 770 |
| Unbounded Granular Material (UGM) | 500 | 200 |

Gauges

**Figure 11.** MLS10 testing system and pavement structure.

The remaining drilling core is used to determine the Young's modulus of the layer where the gauge is glued. After MLS installation and starting cycles, we compared the measured strains from wireless and wired data acquisition systems and numerical values, both measurements are recorded simultaneously. A comparison with our measurement of the longitudinal strain under the bituminous layer is carried out using a semi-analytical calculation model. This numerical tool is presented in the work of Manyo et al. [33]. This multi-layer tire–road contact model has already been validated in the MLS tests. The structure of the modeled pavement is the same as shown in Figure 11. The load is modeled by the contact surface of the real dual tire loaded at 2 × 32.5 kN traveling at a speed of 8 km per hour. Figure 12 shows a comparison of the obtained numerical results and the response of our sensor. This result then confirms the quality of our measurement to find the maximum strain and the shape of the signal.

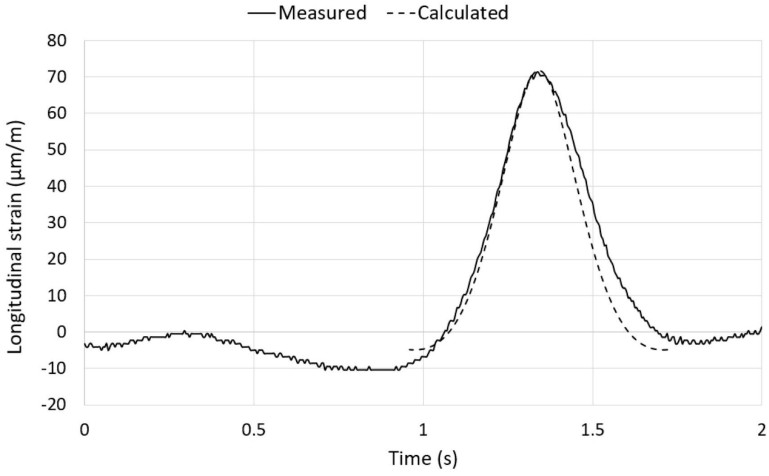

**Figure 12.** Measured and calculated longitudinal strain from in situ pavement testing.

## 5. Results and Discussion

Figure 13 shows the signal measured during the passage of the free-rolling wheels. Simultaneously, we measured the longitudinal strain with a classic wired Wheatstone bridge, and with the wireless measurement channel. Both measurements give the same waveform and the same maximum and minimum values, so we consider this test as conclusive. The waveforms of both measurements are also close to the work of Grellet et al. [11,34]. That confirms the previous test, the electronic design, and the gluing technique. Indeed, Grellet et al. [11] also used a pavement core instrumented with a fiber optic network on a small proof polymeric body. Epoxy glue is used to fix the proof body on the asphalt core, and the latter to the pavement with a glue joint of 1 mm. The described sensor is designed only for the measurement of dynamic load. As a result, it is impossible to determine the long-term strain that appears between the two measurements. As explained in the third section, we use a basic alignment to set the longitudinal gauge axis. To avoid misalignment issues, we might use strain gauge rosettes. With rosettes, we can compute and compensate for misalignment errors; however, it needs to add one measurement channel and this increases the energy consumption and decreases the maximum sampling frequency.

As previously announced, the gauges were chosen as long as possible to measure the average strain directly on asphalt concrete to avoid the use of an intermediate core. The drawback of this technique is the overlapping of longitudinal and transversal gauges. Gauges overlapping decrease the independence between longitudinal and transversal measurement. Indeed, Grellet et al. [11] observed a lower strain than expected with an optical fiber sensor embedded in polymeric body proof due to energy losses occurred by load transfer from the pavement to the sensor. Our choice is a compromise. It might be better if we add a temperature sensor to know the temperature of the gauges and test body.

Indeed, the temperature impact gauges sensibility (around 10% at 40 °C when the gauge is calibrated at 25 °C) and pavement modulus.

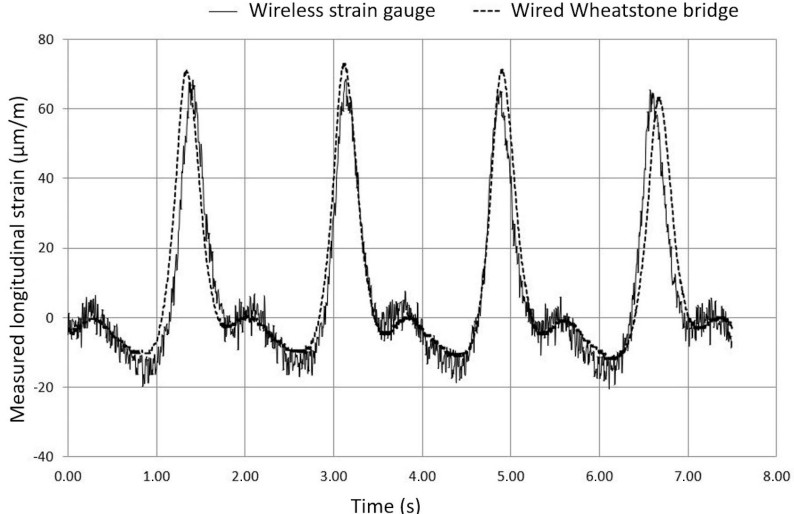

**Figure 13.** Comparison between wireless and Wheatstone bridge measurements.

Concerning the performance and limitations of the electronic section, compromises had also to be made. To save board space and power consumption, we choose to use the built-in voltage reference of MCU to calibrate the ADC. Unfortunately, the output voltage drift due to temperature or supply voltage variation is not specified in the datasheet. As a consequence, the only available information is the output voltage range, which is 1.0 to 1.2 V for a typical value set to 1.1 V. The worst case of voltage reference gives 10% of error, which is the major contribution to inaccuracy. The second contribution is inaccuracy on resistors, which defines the amplifier gain with 2% in the worst case. Concerning gauges, there are the following three parameters: transverse sensitivity (1% of longitudinal sensitivity), gauge factor (2.14 ± 0.5%), and grid resistance (120 Ω ± 0.6%). We neglected the inaccuracy of grid resistance and gauge factor because we calibrated our instrument. It remains sensitive to transverse (that corresponds to ±2 µm/m when the wheel is fully loaded) and gauge factor variation due to temperature (≈0.1% at 50 °C). The absolute accuracy of ADC is ±2 LSB (least significant bit), which corresponds to 2 µm/m. As a result, the sum of the uncertainties of the different stages amounts to the following: 13% and ±4 µm/m. Concerning wireless data transmission, we use a class 1 Bluetooth module that is qualified for IEEE standards 2.1/2.0/1.2/1.1. This part is used to transmit data at 2 Mbps in a 100 m range in free space. In practice, the Bluetooth transceiver is close to the soil and does n0t operate in open fields. As a consequence, the real range is reduced. This consequence did not induce any problems during experiments because the major drawback of wireless transmission with this module is energy consumption and EMI (electromagnetic interference) with analog circuits. Indeed, the wireless transmission absorbs 75% of the total consumption in measurement mode. Also, the wireless signal shown in Figure 13 is noisier than the signal from Figure 12. It is due to the proximity of the transceiver even if this one is certified EN 55022 Class B radiated.

Despite all these limitations, the instrumentation after asphalt layer construction and cooling presented in this article is useful for adapting the test setup, completing a damaged one, or simplifying road instrumentation. The main advantage is the good positioning of the sensor and a reduction in sensors destruction during instrumentation before building the bituminous layer.

### 6. Conclusions

This article introduced the method and feasibility of a deformation measurement device implemented in a core of materials identical to the pavement and re-established by gluing. The device can measure the longitudinal and transverse deformations in a pavement layer. The implementation of the sensor can be carried out after the construction of the asphalt concrete layer. Installation after construction work simplifies the electronic design and the problems of destruction during compaction or high temperatures. The lifespan of sensors installed in roadways is also often limited by the wires. This wireless data transmission technique will therefore increase the lifespan of instrumented structures. The supercapacitor and wireless charger allow the device to be implemented entirely inside the core and remain perfectly autonomous. The electronic part of the wireless sensor is nevertheless more difficult to design than simple wired sensors. Energy storage can, however, increase the size of the circuit and the transceiver can sometimes make noise in the measured signal.

The measurement results were validated using measurements on a conventional gauge system as well as using the results from a reliable numerical model. These main innovations open the way to building long-life real-time pavement monitoring systems. With this system, a sampling frequency of 1 kHz can be achieved, which allows us to capture the dynamic behavior of the pavement (as can be found with higher vehicle speed or a traffic simulator and the falling weight deflectometer test). This high frequency can provide more detailed information about the pavement response to loading, and can help identify critical points within the pavement structure. Finally, we plan for future tests on an instrumented roadway with further testing of this new sensor in different loading, position, and temperature configurations.

**Author Contributions:** Conceptualization, P.R. and C.P.; methodology, C.G.; software, D.C.d.S.; validation, B.P., N.R.-R. and C.P.; formal analysis and investigation, C.G.; data curation, C.G.; writing—original draft preparation, B.P. and C.G.; writing—review and editing, B.P. and N.F.; supervision, C.P. All authors have read and agreed to the published version of the manuscript.

**Funding:** This research received no external funding.

**Institutional Review Board Statement:** Not applicable.

**Informed Consent Statement:** Not applicable.

**Data Availability Statement:** The data presented in this study are available on request from the corresponding author. The data are not publicly available due to privacy.

**Acknowledgments:** The authors would like to acknowledge the company Spie Batignolles Malet (Toulouse, France) and Alain Beghin, for building the pavement layers used for these tests and for his support in developing the test program.

**Conflicts of Interest:** The authors declare no conflicts of interest.

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
