# Peer review of "Wireless Strain Gauge for Monitoring Bituminous Pavements"

_applsci, doi:10.3390/app14062245_

Round 1
Reviewer 1 Report
Comments and Suggestions for Authors
Gillot et al. tested a strain gauge configuration to monitor asphalt roads. The tests are load-independent, which validate the measurements. The described device was tested under a hydraulic press allows measurement and provides reliable results, potentially useful for Structural Health Monitoring of pavement structures. Some suggestions to improve the manuscript are:
minor
How the real strain was calculated in T1 or Fig 11 is unclear and open to interpretation. It seems like the sample used resembles a very thick Brazilian disk, see (Kim 2024). In that case, it introduces a plain strain problem. Please specify.
The equations, or a reference to them (see Freire, 2010), to calculate strains from SG measurements should be included to orient the reader.
major
The abstract seems to cover the key features and benefits of the device, including wireless transmission, rechargeable capability, easy positioning, and the ability to measure deformation when a vehicle passes by. However, it is missing quantitative findings and device limitations.
The literature review does not clearly show the need or the knowledge gap. Please state it clearly.
Is the data transmission and speed affected by external factors such as vibration, temperature, electromagnetic signal, etc? Please state or include them as limitations of the present study.
Please discuss the lag in measuring before ~500N on the loading stage shown in Fig 7.
I do not understand the use of the linear fitting in Fig 8. Please use it or delete it.
***
Kim, S.; Nam, B.H.; Jung, Y.-H. Comparison of Deep Transfer Learning Models for the Quantification of Photoelastic Images. Appl. Sci. 2024, 14, 758. https://doi.org/10.3390/app14020758
Freire, José L.F. EXPERIMENTAL MECHANICS - Electrical Resistance Strain Gages. UNESCO. https://www.eolss.net/Sample-chapters/C05/E6-194-07.pdf
Comments on the Quality of English LanguageT1. "Real and Measured Strain" is not capitalized
Reviewer 2 Report
Comments and Suggestions for Authors
Strain gauge instrumentation for the monitoring of pavements is an issue that has gained a lot of attention in recent years. The authors have developed an instrumentation that appears to have certain advantages over the wired sensors. However, I am not convinced that this set up can be used for the evaluation of pavements at this point. Temperature recording is essential, as mentioned by the authors, for the evaluation of the bituminous layer. Moreover, vehicle speeds (and as such loading frequency) will vary in real conditions.
The following comments need to be addressed.
Are the gauges placed near the top of the asphalt layer or near the bottom of the lower AC layer?
page 10, line 224-225. The thickness and material moduli are not shown on the figure
Reviewer 3 Report
Comments and Suggestions for Authors
This paper proposes a wireless strain gauge for monitoring bituminous pavements. The study is interesting, but not particularly well developed. English need review for the presence of few typos. Also, several points should be clarified before a possible publication.
Major comments:
● Reference in the introduction should be definitely enriched to at least 40-50 citations (16 references is not a satisfactory amount because of the very wide existing literature on the topic related to wireless sensors). Just for suggestion you can cite:
*Di Graziano, A., Marchetta, V., & Cafiso, S. (2020). Structural health monitoring of asphalt pavements using smart sensor networks: A comprehensive review. Journal of Traffic and Transportation Engineering (English Edition), 7(5), 639-651.
*Using commercial UHF-RFID wireless tags to detect structural damage.
● In the introduction, the advantages and, particularly, the drawbacks of using a wireless system should be pointed out
● Figure 7: x-axis and y-axis should be inverted
● A picture of the test body in PVC should be provided while tested with the innovative wireless strain gauge and the traditional one
● Details should be given to support the validation of the new wireless sensor: accuracy precision should be discussed in a dedicated section
● In Figure 7, different curves should be provided (and referred in the legend) for the wireless sensor response and the traditionally used one
Comments on the Quality of English LanguageEnglish is a bit tedious and minor typos are present. Please, check everything including punctuation.
Round 2
Reviewer 1 Report
Comments and Suggestions for Authors
Gillot et al provided with a revised version. The article meets the minimum standards for publication.
Reviewer 3 Report
Comments and Suggestions for Authors
Comments were sufficiently addressed. Paper can be published
Comments on the Quality of English LanguageI did not find typos.